# PyTorch Geometric High Order: A Unified Library for High Order Graph Neural Network

## Abstract

We introduce PyTorch Geometric High Order (PyGHO), a library for High Order Graph Neural Networks (HOGNNs) that extends PyTorch Geometric (PyG). Unlike ordinary Message Passing Neural Networks (MPNNs) that exchange messages between nodes, HOGNNs, encompassing subgraph GNNs and k-WL GNNs, encode node tuples, a method previously lacking a standardized framework and often requiring complex coding. PyGHO's main objective is to provide an unified and user-friendly interface for various HOGNNs. It accomplishes this through streamlined data structures for node tuples, comprehensive data processing utilities, and a flexible suite of operators for high-order GNN methodologies. In this work, we present a detailed in-depth of PyGHO and compare HOGNNs implemented with PyGHO with their official implementation on real-world tasks. PyGHO achieves up to $50\%$ acceleration and reduces the code needed for implementation by an order of magnitude.

## 1 Introduction

Message Passing Neural Networks (MPNNs) (Gilmer et al., 2017) have gained significant popularity across various real-world applications, such as recommender systems (Wu et al., 2022), biology (Zhang et al., 2021), chemistry (Reiser et al., 2022), and combinatorial optimization (Cappart et al., 2023). This widespread adoption owes much to the availability of powerful and versatile MPNN libraries like PyTorch Geometric (PyG) (Fey & Lenssen, 2019), Deep Graph Library (DGL) (Zheng et al., 2020), Spektral (Grattarola & Alippi, 2021), and Jraph (Godwin* et al., 2020).

However, the expressiveness of MPNNs is inherently limited due to their node-level message-passing architecture. To address this limitation and unlock greater expressivity, High Order Graph Neural Networks (HOGNNs) (Zhang & Li, 2021; Zhang et al., 2023; Morris et al., 2019; 2020; Zhao et al., 2022; Bevilacqua et al., 2022; Frasca et al., 2022; Maron et al., 2019a; Huang et al., 2023) have emerged as a promising alternative. Unlike MPNNs, HOGNNs operate by passing messages between node tuples, enabling them to capture higher-order structural information within graphs. This paradigm shift has significantly improved their expressiveness, leading to state-of-the-art performance on various graph-level tasks (Zhang et al., 2023).

However, despite their potential, the implementation of HOGNNs is complex, laborious, and often tied to specific models, hindering their practical adoption in real-world applications. To bridge this gap, we present the PyTorch Geometric High Order (PyGHO) library, the first library for High Order Graph Neural Networks. Our contributions in this work address critical challenges in HOGNN development and application:

- Specialized Data Structures for HOGNNs: Recognizing the need for representing node tuple features, we provide specialized data structures optimized for HOGNNs. These structures enable seamless integration of HOGNN models into real-world applications.

- Data Processing Framework: we offer a user-friendly data processing framework. This framework simplifies data preprocessing and facilitates the smooth incorporation of graph datasets into HOGNN models.

- Flexible Operators on High-Order Tensors: We introduce a comprehensive set of operators for building diverse HOGNNs. This framework not only streamlines the development process but

also serves as a reference point for researchers and practitioners, promoting exploration and advancement in the field of HOGNNs.

In the following sections, we delve into the PyTorch Geometric High Order (PyGHO) library, elucidating its architecture, design principles, and practical applications. By providing an accessible and comprehensive toolkit for high-order graph neural networks, we aim to empower researchers and practitioners to harness the full potential of HOGNNs in solving complex real-world problems.

In the subsequent sections, we delve into the PyTorch Geometric High Order (PyGHO) library, providing insights into its design principles and practical applications. By offering an accessible and comprehensive toolkit for high-order graph neural networks, our aim is to empower researchers and practitioners to fully leverage the potential of HOGNNs in tackling complex real-world challenges.

## 2 RELATED WORK

Several libraries have been developed to facilitate the implementation of Graph Neural Networks (GNNs)(Fey & Lenssen, 2019; Cen et al., 2023; Zheng et al., 2020; Liu et al., 2021; Godwin* et al., 2020; Hu et al., 2021; Grattarola & Alippi, 2021). However, none of these libraries explicitly support high-order GNNs. In response to this gap, our library is designed as an extension of PyTorch Geometric (PyG)(Fey & Lenssen, 2019), one of the leading GNN libraries.

We have strived to maintain a high degree of consistency with PyG's user-facing APIs. This includes preserving the similarity in data preprocessing and loading routines, model definitions, and usage. By doing so, we aim to ensure that users familiar with PyG can seamlessly transition to our library for high-order GNN tasks, simplifying the adoption of these advanced techniques.

## 3 PRELIMINARY

Let $\mathcal{G} = (V, E, X)$ denote a graph with a node set $V = \{1, 2, 3, \ldots, n\}$, an edge set $E \subseteq V \times V$, and a node feature matrix $X \in \mathbb{R}^{n \times d}$. Each row $X_v$ of the matrix corresponds to the features of node $v$. The edge set $E$ can also be represented using the adjacency matrix $A \in \mathbb{R}^{n \times n}$, where the element $(u, v)$ of $A$ is 1 if the edge $(u, v)$ is in $E$ and 0 otherwise. Additionally, the graph $\mathcal{G}$ can be represented by the pair $(A, X)$. Each edge $(i, j)$ may also have a feature $e_{ij}$. Let $N(i) = \{j | (i, j) \in V\}$ denote the neighbor set of node $i$.

Ignoring hidden dimensions, a tensor is considered to be $m$-D if it retains $m$ dimensions whose sizes are associated with properties of the input graph. For instance, if we denote the number of nodes in the input graph as $n$, then the adjacency matrix $\mathbb{R}^{n \times n}$ is 2-D, while the node feature matrix, denoted as $\mathbb{R}^{n \times d}$, is 1-D.

**Message Passing Neural Network (MPNN).** MPNN (Gilmer et al., 2017) is a comprehensive GNN framework that unifies various representative GNNs, including GCN (Kipf & Welling, 2017), GIN (Xu et al., 2019), GraphSage (Hamilton et al., 2017), and GAT (Veličković et al., 2017). It is composed of several message passing layers, each layer updates node representations as follows:

$$h_i^{t+1} \leftarrow U^t(h_i^t, M^t(\{\!\{(h_i^t, h_j^t, e_{ij}) | j \in N(i)\}\!\})), \tag{1}$$

where $h_i^t$ represents the representation of node $v$ at the $t$-th layer, initially set as $h_v^{(0)} = x_i$. $M^t$ represents an aggregation function that maps the multiset of neighbor representations to a single representation, often utilizing operations like summation over the multiset. $U^t$ combines the original node representation with information aggregated from neighbors to produce a new representation. The node representations can subsequently be pooled to generate embeddings for the entire graph:

$$h = P(\{\!\{h_i | i \in V\}\!\}), \tag{2}$$

where $P$ represents a pooling operation, such as summation.

Overall, the message passing framework can be decomposed into operations, like message passing between elements and pooling, performed on the 1-D node representation matrix $h \in \mathbb{R}^{n \times d'}$. Existing libraries like PyTorch Geometric (PyG) (Fey & Lenssen, 2019) provide comprehensive utilities

for these operators, simplifying MPNN development. However, these operators are constrained to 1-D node representation only and are not applicable to high-order representations.

**High-Order GNN (HOGNN).** Unlike MPNN, which focuses on 1-D node representations $h \in \mathbb{R}^{n \times d'}$, High-Order GNNs (Zhang & Li, 2021; Bevilacqua et al., 2022; Zhao et al., 2022; Morris et al., 2019; 2020; Maron et al., 2019a; Zhang et al., 2023; Huang et al., 2023) generate $m$-D tuple representations $H \in \mathbb{R}^{\overbrace{n \times n \times \ldots \times n}^{m} \times d}$, with $m$ is typically 2 or 3, and $d$ is the hidden dimension. While these representations cannot be unified into a single equation like Equation 1, HOGNNs can still be constructed using operations on $m$-D tensors (Frasca et al., 2022; Maron et al., 2019b).

Taking NGNN (with GIN base) (Zhang & Li, 2021) as an example, NGNN first samples a subgraph for each node $i$ and then runs GIN (Xu et al., 2019), a representative MPNN, on all subgraphs simultaneously. It produces a 2-D representation $H \in \mathbb{R}^{n \times n \times d}$, where $H_{ij}$ represents the representation of node $j$ in the subgraph rooted at node $i$. The message passing within all subgraphs can be expressed as:

$$h_{ij}^{t+1} \leftarrow \sum_{k \in N_i(j) \cup \{j\}} \text{MLP}(h_{ik}^t), \tag{3}$$

where $N_i(j)$ represents the set of neighbors of node $j$ in the subgraph rooted at $i$. After several layers of message passing, tuple representations $H$ are pooled to generate the final graph representation:

$$h_i = \text{P}_2 \left( \left\{ h_{ij} | j \in V_i \right\} \right), \quad h_G = \text{P}_1 \left( \left\{ h_i | i \in V \right\} \right), \tag{4}$$

All these modules are composed of operators on high-order tensor, like message passing between tensor elements and pooling. Besides NGNN, other existing subgraph GNNs can also be decomposed into these operators as shown in theory by Zhang et al. (2023). We list how existing HOGNNs can be decomposed into a operators on high-order tensors in implementation in Table 1 of Section 4.3.

## 4 LIBRARY DESIGN

This section provides an overview of the library's design, covering data structures, dataset processing, and operators for High Order Graph Neural Networks (HOGNNs). The library is designed to handle both sparse and dense tensors efficiently.

### 4.1 DATA STRUCTURE

While basic deep learning libraries typically support the high-order tensors directly, HOGNNs demand specialized structures. NGNN, for example, employs a 2-D tensor $H \in \mathbb{R}^{n \times n \times d}$, where $H_{ij}$ represents the node representation of node $j$ in subgraph $i$. Since not all nodes are included in each subgraph, some elements in $H_{ij}$ may not correspond to any node and should not exist. To address this challenge, we introduce two distinct data structures that cater to the unique requirements of HOGNNs: MaskedTensor and SparseTensor.

**MaskedTensor** A MaskedTensor consists of two components: `data`, with shape (`masked shape`, `dense shape`), and `mask`, with shape (`masked shape`). The `mask` tensor contains Boolean values, indicating whether the corresponding element in `data` exists within the tensor. For example, in the context of NGNN's representation $H \in \mathbb{R}^{n \times n \times d}$, `data` resides in $\mathbb{R}^{n \times n \times d}$, and `mask` is in $\{0,1\}^{n \times n}$. Here the `masked shape` is $(n, n)$, and the `dense shape` is $(d)$. The element $(i, j)$ in `mask` is 1 if the tuple $(i, j)$ exists in the tensor. The invalid elements will not affect the output of the operators in this library. For example, the summation over a MaskedTensor will consider the non-existing elements as 0 and thus ignore them.

**SparseTensor** In contrast, SparseTensor stores only existing elements while ignoring non-existing ones. This approach proves to be more efficient when a small ratio of valid elements is present. A SparseTensor, with shape (`sparse shape`, `dense shape`), comprises two tensors: `indices` (an Integer Tensor with shape (`sparse dim, nnz`)) and `values` (with shape (`nnz, dense shape`)). Here, `sparse dim` represents the number of dimensions in the sparse shape, and `nnz` stands for the count of existing elements. The columns of `indices` and rows of `values` correspond to the non-zero elements, making it straightforward to retrieve and manipulate the required information. For example, in NGNN's representation $H \in \mathbb{R}^{n \times n \times d}$, assuming the total number of nodes in subgraphs is $m$, $H$ can be represented as `indices` $a \in \mathbb{N}^{2 \times m}$ and `values` $v \in \mathbb{R}^{m \times d}$. Here `sparse shape` is $(n, n)$, `sparse dim` is 2, and `dense shape` is $(d)$. Specifically, for $i = 1, 2, \ldots, n$, $H_{a_{1,i}, a_{2,i}} = v_i$.

The foundational concepts of these data structures have been used in various implementations, but our implementation remains unique. For example, NGNN (Zhang & Li, 2021) utilizes integer tensors as indices and value tensors for tuple representations. However, these implementations haven't integrated these code components into a unified data structure with a consistent API. This fragmented implementation often makes the codebase hard to understand or extend. While PyTorch and PyG also provide sparse tensors API, they don't support operators in our library, like message passing and pooling. Additionally, these prior works primarily use either sparse or dense data structures exclusively, limiting their versatility. In contrast, our framework offers both routines and facilitates the application of the same model architecture to both sparse and dense representations (see Section 4.3).

## 4.2 HIGH ORDER GRAPH DATA PROCESSING

HOGNNs and ordinary MPNNs share graph tasks, allowing us to reuse PyTorch Geometric's (PyG) data processing routines. However, due to the specific requirements for precomputing and preserving high-order features, we have significantly extended these routines within PyGHO. As a result, PyGHO's data processing capabilities remain highly compatible with PyG while offering convenience for HOGNNs. To illustrate, NGNN (Zhang & Li, 2021)'s official code uses more than 600 lines for dataset processing, while with PyGHO, the same utilities can be implemented within 8 lines. The framework is also highly flexible allowing using custom tuple samplers and features.

**High Order Feature Precomputation** High-order feature precomputation can be efficiently conducted in parallel using the PyGHO library. To illustrate, consider the following example:

```
# Ordinary PyG dataset
trn_dataset = ZINC("dataset/ZINC", subset=True, split="train")
# High-order graph dataset
trn_dataset = ParallelPreprocessDataset("dataset/ZINC_trn", trn_dataset,
                            pre_transform=Sppretransform(
                            tuplesamplers=partial(KhopSampler,
                            hop=3)), num_workers=8)
```

The `ParallelPreprocessDataset` class takes an ordinary PyG dataset as input and performs transformations on each graph in parallel (utilizing 8 processes in this example). Here, the `tuplesamplers` parameter represents functions that take a graph as input and produce a sparse tensor. Multiple samplers can be applied simultaneously, and the resulting output is assigned the names specified in the `annotate` parameter. As an example, we use `partial(KhopSampler, hop=3)`, a sampler designed for NGNN, to sample a 3-hop ego-network rooted at each node. The shortest path distance to the root node serves as the tuple features. The produced SparseTensor is then saved and can be effectively used to initialize tuple representations.

Since the dataset preprocessing routine is closely related to data structures, we have designed two separate routines for sparse and dense tensor, which only differ in the `pre_transform` function. For dense tensors, we can simply use `Mapretransform(None, tuplesamplers)`. In this case, the `tuplesamplers` is a list of functions that produce a dense high-order MaskedTensor containing tuple features. For both sparse and dense data preprocessing, the tuplesampler can be custom function.

**Mini-Batch and Data Loader**   Enabling batch training in HOGNNs requires handling graphs of varying sizes, which is not a trivial task. Different strategies are employed for sparse and masked tensor data structures.

For sparse tensor data, the solution is relatively straightforward. We can concatenate the tensors of each graph along the diagonal of a larger tensor: For instance, in a batch of $B$ graphs with adjacency matrices $A_i \in \mathbb{R}^{n_i \times n_i}$, node features $x \in \mathbb{R}^{n_i \times d}$, and tuple features $X \in \mathbb{R}^{n_i \times n_i \times d'}$ for $i = 1, 2, \ldots, B$, the features for the entire batch are represented as $A \in \mathbb{R}^{n \times n}$, $x \in \mathbb{R}^{n \times d}$, and $X \in \mathbb{R}^{n \times n \times d'}$, where $n = \sum_{i=1}^{B} n_i$. The concatenation is as follows,

$$
A = \begin{bmatrix} A_1 & 0 & 0 & \cdots & 0 \\ 0 & A_2 & 0 & \cdots & 0 \\ 0 & 0 & A_3 & \cdots & 0 \\ \vdots & \vdots & \vdots & \vdots & \vdots \\ 0 & 0 & 0 & \cdots & A_B \end{bmatrix}, x = \begin{bmatrix} x_1 \\ x_2 \\ x_3 \\ \vdots \\ x_B \end{bmatrix}, X = \begin{bmatrix} X_1 & 0 & 0 & \cdots & 0 \\ 0 & X_2 & 0 & \cdots & 0 \\ 0 & 0 & X_3 & \cdots & 0 \\ \vdots & \vdots & \vdots & \vdots & \vdots \\ 0 & 0 & 0 & \cdots & X_B \end{bmatrix} \tag{5}
$$

This arrangement allows tensors in batched data have the same number of dimension as those of a single graph and thus share common operators. We provides PygHO's own dataloader. It has the compatible parameters to PyTorch's DataLoader and further do concatenations above.

```
from pygho.subgdata import SpDataloader
trn_dataloader = SpDataloader(trn_dataset, batch_size=32, shuffle=True,
                              drop_last=True)
```

Here, similar to vanilla PyTorch dataloader, PyGHO's dataloader takes a datasets and hyperparameters like `batch_size`, `shuffle`, `drop_last` as input.

As concatenation along the diagonal leads to a lot of non-existing elements, handling Masked Tensor data involves a different strategy for saving space. In this case, tensors are padded to the same shape and stacked along a new axis. For example, in a batch of $B$ graphs with adjacency matrices $A_i \in \mathbb{R}^{n_i \times n_i}$, node features $x \in \mathbb{R}^{n_i \times d}$, and tuple features $X \in \mathbb{R}^{n_i \times n_i \times d'}$ for $i = 1, 2, \ldots, B$, the features for the entire batch are represented as $A \in \mathbb{R}^{B \times \tilde{n} \times \tilde{n}}$, $x \in \mathbb{R}^{B \times \tilde{n} \times d}$, and $X \in \mathbb{R}^{B \times \tilde{n} \times \tilde{n} \times d'}$, where $\tilde{n} = \max\{n_i | i = 1, 2, \ldots, B\}$.

$$
A = \begin{bmatrix} \begin{pmatrix} A_1 & 0_{n_1, \tilde{n}-n_1} \\ 0_{\tilde{n}-n_1, n_1} & 0_{n_1, n_1} \end{pmatrix} \\ \begin{pmatrix} A_2 & 0_{n_2, \tilde{n}-n_2} \\ 0_{\tilde{n}-n_2, n_2} & 0_{n_2, n_2} \end{pmatrix} \\ \vdots \\ \begin{pmatrix} A_B & 0_{n_B, \tilde{n}-n_B} \\ 0_{\tilde{n}-n_B, n_B} & 0_{n_B, n_B} \end{pmatrix} \end{bmatrix}, x = \begin{bmatrix} \begin{pmatrix} x_1 \\ 0_{\tilde{n}-n_1, d} \end{pmatrix} \\ \begin{pmatrix} x_2 \\ 0_{\tilde{n}-n_2, d} \end{pmatrix} \\ \vdots \\ \begin{pmatrix} x_B \\ 0_{\tilde{n}-n_B, d} \end{pmatrix} \end{bmatrix}, X = \begin{bmatrix} \begin{pmatrix} X_1 & 0_{n_1, \tilde{n}-n_1} \\ 0_{\tilde{n}-n_1, n_1} & 0_{n_1, n_1} \end{pmatrix} \\ \begin{pmatrix} X_2 & 0_{n_2, \tilde{n}-n_2} \\ 0_{\tilde{n}-n_2, n_2} & 0_{n_2, n_2} \end{pmatrix} \\ \vdots \\ \begin{pmatrix} X_B & 0_{n_B, \tilde{n}-n_B} \\ 0_{\tilde{n}-n_B, n_B} & 0_{n_B, n_B} \end{pmatrix} \end{bmatrix} \tag{6}
$$

This padding and stacking strategy ensures consistent shapes across tensors, allowing for efficient processing of dense data. We also provide the dataloader to implement it conveniently.

```
from pygho.subgdata import MaDataloader
trn_dataloader = MaDataloader(trn_dataset, batch_size=256, device=device,
                              shuffle=True, drop_last=True)
```

## 4.3 OPERATORS

In the preceding sections, we have introduced data structures tailored to the representation of high-order Graph Neural Networks (HOGNNs) along with a novel data processing routine. Therefore, the learning methodologies within HOGNNs can be deconstructed into operations executed on these tensor structures. Our comprehensive codebase for these operations has been thoughtfully structured into three distinct layers, each contributing to the versatility and utility of our library:

**Layer 1: Backend** The `pygho.backend` module serves as the foundation, encompassing fundamental data structures and their associated operations. In this layer, the emphasis lies on tensor operations, without delving into the settings of graph learning. The key components encapsulated within this layer encompass:

- Matrix Multiplication: This method equips users with versatile matrix multiplication capabilities, accommodating scenarios involving two SparseTensors, one sparse and one MaskedTensor, and two MaskedTensors. It also seamlessly handles batched matrix multiplication. Furthermore, it offers alternative operations, allowing for the replacement of the standard summation with maximum and mean calculations. For sparse tensors, it additionally incorporates a generalized matrix multiplication, facilitating a wide range of message passing operations.
- Matrix Addition: This operation provides the means to add two matrices, whether they are sparse or dense, enabling flexible manipulation of data.
- Reduce Operations: A suite of essential reduction operations is included, encompassing sum, mean, max, and min, designed to effectively collapse dimensions within tensors.
- Expand Operation: This operation enables the augmentation of tensors by introducing new dimensions.
- Tuplewise Apply (func): A utility function that systematically applies a user-specified function to each individual element within the tensor, facilitating custom tuple-wise transformations.
- Diagonal Apply (func): This operation is tailored for applying a given function to the diagonal elements of tensors, offering specialized processing capabilities for transformations on diagonal elements.

**Layer 2: Graph Operators.** Layer 2 builds upon Layer 1 and consists of the `pygho.honn.SpOperator`, `pygho.honn.MaOperator` modules, engineered for graph operations involving SparseTensor and MaskedTensor structures. Furthermore, the `pygho.honn.TensorOp` layer acts as an encompassing wrapper for these operators, seamlessly abstracting away the nuances between Sparse and Masked Tensor data structures. Within this layer, we encounter an array of operations, each designed to facilitate diverse aspects of graph processing:

- This operation enables the seamless transmission of messages between tuples of nodes. To illustrate, consider the message passing operation within each subgraph simultaneously (like NGNN (Zhang & Li, 2021)), as exemplified by the equation:

$$h_{ij} \leftarrow \sum_{k \in N_i(j)} h'_{ik}, \qquad (7)$$

This can be readily implemented using the message passing operator with tuple representation $H'$ and adjacency matrix $A$ as input:

$$H \leftarrow H'A \qquad (8)$$

While this transformation may appear straightforward, several details deserve attention:

- Optimization for Induced Subgraph Input: In the original Equation 7, the summation is confined to neighbors within the subgraph. However, the matrix multiplication in Equation 8 seemingly extends beyond the subgraph to include neighbors outside of it. In fact, this apparent inconsistency is a non-issue, as the subgraphs in NGNN are inherently induced by a specific node set. Consequently, any neighbors located outside the subgraph are automatically treated as non-existent and have no bearing on the final result. Importantly, our implementation has been designed to optimize for scenarios involving induced subgraphs. In cases that the subgraphs are not induced by a node set, pygho also provides operators designed to handle such situations.
- Optimization for Sparse Output: The operation $H'A$ may generate non-zero elements for pairs $(i, j)$ that do not exist in the subgraph. To enhance efficiency for sparse input tensors $H$ and $A$, we've optimized the multiplication to prevent the computation of such non-existent elements.
- Aggregator Beyond Sum: Not all aggregators utilized in GNNs can be expressed through simple matrix multiplication, such as the attention aggregation employed in the Graph Attention Network (GAT). However, our message passing operator accommodates custom aggregators, offering versatility in modeling.

- Message Passing Across Subgraphs: The operation $H'A$ facilitates message passing within each subgraph, enabling the exchange of representations between nodes within the same subgraph. Furthermore, we extend support to the message operator at other dimensions, which fosters message exchange between identical nodes situated in different subgraphs. In fact, pygho furnishes operators for arbitrary dimensions, catering to high-order tensor requirements.

- Pooling: This operation serves to reduce high-order tensors to lower-order counterparts, achieved through summation, maximum value extraction, or mean computation across specified dimensions. For instance, in a 2D-representation $H \in \mathbb{R}^{n \times n \times d}$, dimension 0 pooling consolidates representations of the same node in distinct subgraphs into node presentations, while dimension 1 pooling combines representations of nodes within the same subgraph into subgraph presentations.

- Diagonal: The diagonal operation reduces high-order tensors to lower-order ones by extracting diagonal elements. For example, given a 2D-representation $H \in \mathbb{R}^{n \times n \times d}$, $\text{diag}(H) \in \mathbb{R}^{n \times d}$ yields the representation of each subgraph's root node.

- Unpooling: This operation performs the reverse of pooling, expanding low-order tensors to high-order ones. For instance, with a node representation $h \in \mathbb{R}^{n \times d}$, dimension 0 unpooling corresponds to assigning $x_i$ to node $i$ across different subgraphs, while dimension 1 unpooling pools the representation of node $i$ across all nodes within the subgraph rooted at $i$.

**Layer 3: Models** Based on Layer 1 and Layer 2, Layer 3 is a repository of distinguished high-order Graph Neural Network (HOGNN) layers, which encompass NGNN (Zhang & Li, 2021), GNNAK (Zhao et al., 2022), DSSGNN (Bevilacqua et al., 2022), SUN (Frasca et al., 2022), SSWL (Zhang et al., 2023), PPGN (Maron et al., 2019a), and I$^2$-GNN. These layers are the fusion of tuplewise neural network transformations and graph operators. The tuplewise neural network transformation can be effortlessly implemented as follows:

```
X = X.tuplewiseapply(MLP)
```

Existing models can be implemented with graph operators as in shown Table 1. In the original implementations, crafting these message passing schemes typically demands hundreds of lines of code. In contrast, our approach stands out by requiring at most 30 lines, showcasing a substantial reduction in code complexity. These examples serve as a compelling testament to how our library's operators significantly enhance flexibility in model development.

Table 1: Graph Operators in Models. Here, $M_i$, $P_i$, and $U_i$ denote message passing operators, pooling operators, and unpooling operators in the $i$-th dimension. $D$ signifies the diagonal operator. $\circ$ represents functional composition.

| Model | Representation | Operator |
|---|---|---|
| NGNN | 2D, Sparse | $M_1$ |
| GNNAK | 2D, Sparse | $M_1, U_0 \circ P_0, U_1 \circ P_1$ |
| DSSGNN | 2D, Sparse | $M_1, U_1 \circ M_0 \circ P_1$ |
| SUN | 2D, Sparse | $M_1, U_0 \circ D, U_1 \circ D, U_0 \circ P_0,$ $U_1 \circ P_1, U_0 \circ M_0 \circ P_0$ |
| I2GNN | 3D, Sparse | $M_2$ |
| DRFWL | 2D, Sparse | $M_1$ |
| SSWL | 2D, Dense | $M_1, M_0$ |
| PPGN | 2D, Dense | $M_1$ |

Furthermore, it's worth noting that existing models are often constrained by their implementation based on either sparse or dense representations. However, our library empowers users to seamlessly leverage the same architecture for both sparse and dense inputs, unlocking newfound versatility in model design and deployment.

## 5 EXPERIMENTS

In our experiments, we implemented a diverse selection of pre-existing models using PyGHO on the ZINC datasets and conducted a meticulous comparative analysis against their official implementations. The experimental details are shown in Appendix A. The results of these experiments are thoughtfully presented in Table 2. Impressively, PyGHO's implementations consistently outperform or rival the official implementations, underscoring its efficacy. Moreover, PyGHO delivers a substantial reduction in execution time, with the SSWL model, in particular, demonstrating an impressive 50% reduction. Equally noteworthy is the remarkable reduction in the lines of code

Table 2: Implementation of existing models with PyGHO on ZINC datasets. The reported metrics include the test Mean Average Error (MAE), time per epoch, and GPU memory consumption during training with a fixed batch size of 128, while keeping the number of layers and hidden dimensions consistent. Furthermore, the table shows the number of lines of code required for both model development (Model column, including the definition of corresponding models in PyGHO) and data processing (Data column). Notably, (P) indicates PyGHO's implementation, while (O) signifies the official implementations. As DRFWL has not released the official code and NGAT is a newly proposed models, we leave the corresponding cells to blank.

|  | MAE (P) | MAE (O) | time /s (P) | time /s (O) | Mem /GB (P) | Mem /GB (O) | Model (P) | Model (O) | Data (P) | Data (O) |
|---|---|---|---|---|---|---|---|---|---|---|
| NGAT | $0.077_{\pm 0.005}$ | - | 5.95 | - | 1.33 | - | 102 | - | 20 | - |
| NGNN | $0.082_{\pm 0.003}$ | $0.111_{\pm 0.003}$ | 5.14 | 6.44 | 1.27 | 1.20 | 97 | 140 | 20 | 694 |
| GNNAK | $0.084_{\pm 0.006}$ | $0.080_{\pm 0.001}$ | 9.49 | 13.03 | 2.48 | 2.36 | 117 | 525 | 20 | 221 |
| DSSGNN | $0.080_{\pm 0.004}$ | $0.102_{\pm 0.003}$ | 6.53 | 9.27 | 1.78 | 1.69 | 107 | 433 | 20 | 412 |
| SUN | $0.094_{\pm 0.001}$ | $0.083_{\pm 0.003}$ | 20.15 | 20.93 | 2.47 | 3.72 | 119 | 777 | 20 | 846 |
| DRFWL | $0.082_{\pm 0.005}$ | $0.077_{\pm 0.002}$ | 7.16 | - | 3.29 | - | 98 | - | 20 | - |
| I2GNN | $0.084_{\pm 0.004}$ | $0.083_{\pm 0.001}$ | 9.41 | 14.92 | 4.74 | 2.5 | 107 | 1048 | 20 | 1384 |
| SSWL | $0.085_{\pm 0.002}$ | $0.083_{\pm 0.003}$ | 19.43 | 45.30 | 9.92 | 3.89 | 103 | 181 | 20 | 58 |
| PPGN | $0.079_{\pm 0.001}$ | $0.079_{\pm 0.005}$ | 13.57 | 20.21 | 7.76 | 20.37 | 98 | 246 | 20 | 285 |

required, often by an order of magnitude, showcasing PyGHO's ability to simplify and streamline complex implementations.

In addition to replicating existing models, we introduce a novel model called NGAT to exemplify the remarkable flexibility of our library. NGAT, akin to NGNN, runs a Graph Neural Network (GNN) on each subgraph concurrently. However, NGAT incorporates attention mechanisms, traditionally challenging to implement in prior works but effortlessly achievable with PyGHO. Notably, NGAT exhibits superior performance compared to NGNN, illuminating PyGHO's potential to catalyze the development of innovative models in the field of graph-based machine learning.

## 6 Conclusion

PyTorch Geometric High Order (PyGHO) is introduced as a unified library for High Order Graph Neural Networks (HOGNNs), seamlessly extending PyTorch Geometric (PyG). PyGHO offers fundamental data structures, efficient data processing interfaces, and abstracted operators, reducing both implementation complexity and execution times in experimental settings. This versatile library is poised to accelerate the development of innovative HOGNN methods and broaden the application scope of HOGNNs across diverse downstream tasks, further enriching the landscape of graph-based machine learning.

## 7 Limitation

We have only conducted benchmarking on the ZINC dataset. In future work, we plan to expand our testing to include a broader range of datasets and tasks. Additionally, we aim to comprehensively explore the design space of HOGNNs, conducting comparisons among various model designs, initial tuple sampling strategies, and pooling techniques.

## 8 Reproducibility Statement

Our code is in the supplementary material.

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

## A  EXPERIMENTAL DETAIL

ZINC (Gómez-Bombarelli et al., 2016) is a dataset of 12000 small molecules. The split is provided by the original lease, with 10000/1000/1000 graphs for training/validation/test, respectively. The target to is to predict the constrained solubility of the whole graph.

NGAT is a model proposed on our own with code. We run all experiments on a linux server with NVIDIA RTX 3090 GPU. All models uses 6 layers, hidden dimension 128, , SiLU activation function, and BatchNorm. They are optimized with Adam optimizer and cosannealing scheduler. More details of the hyperparameters are shown in our code in attachment.

