# OpenReview forum: "PyTorch Geometric High Order: A Unified Library for High Order Graph Neural Network"
_ICLR.cc/2024/Conference — Submitted to ICLR 2024_

### Official Review · Reviewer_nV5W · 2023-10-28

**Soundness:** 2 fair
**Presentation:** 2 fair
**Contribution:** 2 fair
**Rating:** 5
**Confidence:** 3

**Summary:**

The paper is an in-depth overview of PyTorch Geometric High Order (PyGHO), a library that simplifies the implementation of High Order Graph Neural Networks (HOGNNs) in PyTorch Geometric (PyG). The paper provides insights into the architecture, design principles, and practical applications of PyGHO, and compares its performance with official implementations on real-world tasks. The paper aims to empower researchers and practitioners to fully leverage the potential of HOGNNs in tackling complex real-world challenges.

**Strengths:**

1. Comprehensive overview: The paper provides a comprehensive overview of PyTorch Geometric High Order (PyGHO) library, which simplifies the implementation of High Order Graph Neural Networks (HOGNNs) in PyTorch Geometric (PyG).
2. Insights into architecture and design principles: The paper provides insights into the architecture, design principles, and practical applications of PyGHO, which can help researchers and practitioners better understand how to use the library.
3. Real-world performance comparison: The paper compares the performance of PyGHO with official implementations on real-world tasks, which can help researchers and practitioners make informed decisions about which library to use for their specific needs.
4. Empowering researchers and practitioners: The paper aims to empower researchers and practitioners to fully leverage the potential of HOGNNs in tackling complex real-world challenges, which can have a significant impact on the field of AI and machine learning.

**Weaknesses:**

•	This article is more like an engineering implementation and lacks enough innovation.
•	The experiments are limited in scope - PyGHO is only evaluated on the ZINC molecular dataset for graph property prediction. Testing on more diverse datasets and tasks could better demonstrate its general utility.
•	There is no ablation study on design choices like different tuple sampling strategies or pooling techniques. Exploring the large design space of HOGNNs enabled by PyGHO could provide more insights.
•	Scalability to large graphs is not explored. The focus is on small molecular datasets. Ability to handle large real-world networks is unclear.
•	Adoption and use in downstream applications is not shown. Impact on advancing HOGNN research remains to be demonstrated.
•	There are limited optimizations implemented for sparse data structures. Additional optimizations could improve efficiency further.

**Questions:**

See Weaknesses

---

### Official Review · Reviewer_skCY · 2023-10-31

**Soundness:** 3 good
**Presentation:** 3 good
**Contribution:** 2 fair
**Rating:** 5
**Confidence:** 3

**Summary:**

This paper introduces a new toolbox designed for efficiently implementing a higher-order Message-passing scheme based on PyG.

**Strengths:**

The paper makes a significant contribution to the efficient implementation of effective higher-order GNNs. The engineering contribution is substantial and noteworthy.

**Weaknesses:**

The paper could benefit from a comparative analysis between the proposed PyGHO and existing code libraries. The paper restructures the data structure, data loader, and operators in higher-order GNNs, all of which can be achieved with code in PyG. It would be beneficial to provide comparisons of these three components, either empirically or theoretically, between PyGHO and existing methods. The paper provides an overall evaluation of time/memory, which seems somewhat insufficient compared to existing libraries.

**Questions:**

Please check the weakness.

---

### Official Review · Reviewer_gNYJ · 2023-11-01

**Soundness:** 2 fair
**Presentation:** 1 poor
**Contribution:** 2 fair
**Rating:** 3
**Confidence:** 5

**Summary:**

The paper introduces PyTorch Geometric High Order (PyGHO), a library for High Order Graph Neural Networks (HOGNNs) that extends PyTorch Geometric (PyG).
An open-source library regarding high-order graphs is a good contribution to the community.

**Strengths:**

An open-source library regarding high-order graphs is a good contribution to the community.

**Weaknesses:**

Overall, while providing an open-source library is a good contribution to the graph machine learning community, the current state of the paper should be better considered as a workshop submission, rather than a main conference paper submission.
- The paper is obviously below the page limit of 8 pages. It does not seem to be fully finished by the authors.
- The paper provides little technical novelty
- Comparisons with the standard PyG library on standard graph datasets are needed.

**Questions:**

How does PyGHO compare with the standard PyG library on standard graph datasets?

---

### Official Review · Reviewer_tzV7 · 2023-11-05

**Soundness:** 2 fair
**Presentation:** 2 fair
**Contribution:** 2 fair
**Rating:** 3
**Confidence:** 3

**Summary:**

In this paper, the authors propose a new library, PyGHO, designed for High Order Graph Neural Networks (HOGNNs) based on PyTorch Geometric. The library aims to provide an intuitive and user-friendly interface to facilitate HOGNN research. Some experiments are conducted to compare the efficiency and performance between implementations using different libraries.

**Strengths:**

1. It's good to see a new framework with a different angle to facilitate the research in graph learning community
2. The descriptions of the operators are sufficient.

**Weaknesses:**

1. The motivation for designing a new library specifically for HOGNN is not very clear to me. The main claim in the paper is that HOGNN usually generates 2-D (or more) representations which cannot be naturally handled by PyG. But my question is why do we have to naturally support this feature (m-D tuple representations)? Using the example NGNN in the paper, NGNN has a 2-D tensor $H \in \mathbb{R}^{n\times n \times d}$ where $H_{ij}$ represents the node representation of node $j$ in the subgraph $i$. In this case, I am curious why we cannot directly use a list of 1-D tensors where each item contains node representations in a subgraph. If we use this naive implementation/design, are there any disadvantages? In my view, this is the core/fundamental question for designing a new library and it should be clarified clearly. Otherwise, it is a bit difficult to understand the motivation.
2. The empirical experimental results are not very informative. In the results, the authors show that the time spent using the proposed PyGHO is less than the time using the original implementation. But there is no detailed analysis for this. I wonder where the efficiency improvement comes from. I believe that the profiling is needed to have a deeper investigation.

**Questions:**

See the questions in weaknesses.

---

### Meta-Review · Area_Chair_rTEk · 2023-12-10

**Metareview:**

The authors propose a new software library, PyGHO, designed for High Order Graph Neural Networks.

While the software contribution is welcome, there is consensus among the reviewers that the novelty and the comparison with the existing libraries is limited, for a full-fledge conference submission. The authors could consider submitting to a workshop or to a ML software venue.

**Justification For Why Not Higher Score:**

There is consensus among the reviewers that the novelty and the comparison with the existing libraries is limited, for a full-fledge conference submission.

**Justification For Why Not Lower Score:**

N/A

---

### Decision · Program_Chairs · 2024-01-16

Reject